# How Infection and Vaccination Are Linked to Acute and Chronic Urticaria: A Special Focus on COVID-19

**DOI:** 10.3390/v15071585

**Published:** 2023-07-20

**Authors:** Emek Kocatürk, Melba Muñoz, Daniel Elieh-Ali-Komi, Paulo Ricardo Criado, Jonny Peter, Pavel Kolkhir, Pelin Can, Bettina Wedi, Michael Rudenko, Maia Gotua, Luis Felipe Ensina, Clive Grattan, Marcus Maurer

**Affiliations:** 1Institute of Allergology, Charité—Universitätsmedizin Berlin, Corporate Member of Freie Universität Berlin and Humboldt-Universität zu Berlin, 12203 Berlin, Germany; 2Allergology and Immunology, Fraunhofer Institute for Translational Medicine and Pharmacology ITMP, 12203 Berlin, Germanydaniel.elieh-ali-komi@charite.de (D.E.-A.-K.); 3Department of Dermatology, Koç University School of Medicine, Istanbul 34010, Turkey; 4Department of Dermatology, School of Medicine, Centro Universitário Faculdade de Medicina do ABC (CUFMABC), Santo André 09060-870, Brazil; 5Lung Institute, Division of Allergy and Clinical Immunology, Groote Schuur Hospital, University of Cape Town, Cape Town 7925, South Africa; 6Department of Dermatology, Bahçeşehir University, Istanbul 34070, Turkey; pulkumen@gmail.com; 7Department of Dermatology and Allergy, Comprehensive Allergy Center, Hannover Medical School, 30625 Hannover, Germany; 8London Allergy and Immunology Centre, London W1G 9QD, UK; 9Center of Allergy and Immunology, David Tvildiani Medical University, Tbilisi 0159, Georgia; 10Division of Allergy, Clinical Immunology and Rheumatology, Department of Pediatrics, Federal University of São Paulo, São Paulo 01308-000, Brazil; 11Guy’s Hospital, St John’s Institute of Dermatology, London SE1 7EP, UK

**Keywords:** urticaria, angioedema, viral infections, COVID-19, SARS-CoV-2, omalizumab, vaccine, vaccination, adverse events, exacerbation, anaphylaxis

## Abstract

Since more than a century ago, there has been awareness of the connection between viral infections and the onset and exacerbation of urticaria. Our knowledge about the role of viral infection and vaccination in acute and chronic urticaria improved as a result of the COVID-19 pandemic but it has also highlighted knowledge gaps. Viral infections, especially respiratory tract infections like COVID-19, can trigger the onset of acute urticaria (AU) and the exacerbation of chronic urticaria (CU). Less frequently, vaccination against viruses including SARS-CoV-2 can also lead to new onset urticaria as well as worsening of CU in minority. Here, with a particular focus on COVID-19, we review what is known about the role of viral infections and vaccinations as triggers and causes of acute and chronic urticaria. We also discuss possible mechanistic pathways and outline the unmet needs in our knowledge. Although the underlying mechanisms are not clearly understood, it is believed that viral signals, medications, and stress can activate skin mast cells (MCs). Further studies are needed to fully understand the relevance of viral infections and vaccinations in acute and chronic urticaria and to better clarify causal pathways.

## 1. Introduction

Urticaria is a common mast cell-driven disease characterized by the sudden appearance of pruritic wheals and/or angioedema. It is classified as chronic urticaria (CU) when it lasts longer than six weeks, as chronic spontaneous urticaria (CSU) when specific triggers are not known, and chronic inducible urticaria (CIndU) when signs and symptoms develop after definite triggers [1,2]. In all types of urticaria, the release of histamine and other mediators from skin MCs leads to the formation of wheals in the superficial dermis and angioedema in the deeper dermis or subcutaneous tissue [3]. In CIndU, the reasons for the activation of MCs and basophils remain largely unknown. In CSU, there is ample and growing evidence that autoantibodies contribute to the pathogenesis [4]. The two distinct autoimmune responses that have been suggested to be relevant in CSU are (1) type I autoimmunity (autoallergy), which entails the occurrence of IgE autoantibodies to self-antigens such as thyroid peroxidase, and (2) type IIb autoimmunity, which entails IgG autoantibodies against the high-affinity IgE receptor, FcεR1, or IgE itself [4] (Figure 1). Stress, hormones, infections, and pseudoallergens are thought to be triggering factors rather than actual causes of CSU [4]. The role of H. pylori infection, bacterial infections, and bowel parasites in urticaria is not clear; however, the urticaria treatment guideline suggests the elimination of them if identified [1]. In some cases, the onset of urticaria has been linked to viral infection. Viral infection may trigger an immune response in the skin that results in persistently increased cutaneous MC “releasability” [5]. Viral infections are believed to be one of the leading causes of AU in children [5] followed by medication, food, and insect sting reactions [6]. The most common viral infections implicated to drive the onset of AU are upper respiratory tract and gastrointestinal tract infections [7]. COVID-19 is no exception to these, as expected, and numerous cases of AU following COVID-19 have been reported, mostly occurring concurrently with the infection or proceeding the infection in a minority of cases [8]. While there is scarce information in the literature on whether viral infections including COVID-19 can induce AU that becomes chronic, a recent case series described features of CSU triggered by COVID-19 [9]. Another aspect of this concept is that viral infections including COVID-19 can cause exacerbations of CU [10].

Many CU patients and their attending physicians were concerned about the impact of COVID-19 on the development and treatment of CU during the pandemic. A study (COVID-CU) was conducted by Urticaria Centers for Reference and Excellence (UCARE) [11] to determine how the pandemic affected CU-treating physicians and CU patients and also how patients get affected when they catch COVID-19 [10]. This study demonstrated how the pandemic significantly hampered the care provided to CU patients, as seen by the weekly drop in the number of CU patients treated at UCAREs, a significant decline in in-person consultations, and an increase in remote consultations. Systemic corticosteroids and cyclosporine were used less frequently because of the pandemic; however, antihistamine and omalizumab therapies were not significantly impacted [10]. The study, which also investigated the impact of COVID-19 on CU and vice versa also showed that having CU does not increase the risk of COVID-19 becoming severe but that at least a third of CU patients experience worsening of their urticaria with SARS-CoV-2 infection [10].

During the mass vaccination campaign against COVID-19 at the end of 2020, many CU patients were concerned about getting vaccinated as they were frightened of adverse reactions or urticaria exacerbations. The COVAC-CU project was initiated by the UCARE network to evaluate whether the COVID-19 vaccine increases the risk of severe allergic responses and/or urticaria exacerbations in patients with CU and to identify risk factors associated with these reactions. This study showed that COVID-19 vaccinations are safe in CU patients and that 9% and 0.25% of patients experience an exacerbation of their urticaria and systemic allergic reactions, respectively [12].

In this review, we discuss the role, significance, and potential mechanisms of viral infections, such as COVID-19, as well as vaccines that can cause AU and exacerbate CU. We also address unmet needs and future directions on this topic.

## 2. The Role of Viral Infections Including COVID-19 in Causing Acute Urticaria

### 2.1. How Often Is Acute Urticaria Caused by a Viral Infection; How Often Does a Viral Infection Lead to Acute Urticaria?

MC-orchestrated sentinel responses to pathogens, including certain viruses, are generally protective but can lead to enhanced inflammation or autoimmunity. Viral infections may contribute to the development of AU which can then become chronic in 5–39% of the cases with possibly circulating autoantibodies [13,14,15,16]. Magen E et al. supported this idea with their study, showing that 36% of AU cases turned to CSU at week 7 and that these patients were characterized by a positive autologous serum skin test (ASST), thyroid autoimmunity, and profound basopenia at the baseline [17].

AU is a prevalent condition thought to affect 12–23.5% of the general population at least once in their lifetime [18,19] with a cumulative incidence of 18.7% [20]. While 30–50% of cases of AU are considered to be idiopathic, the main proposed causes are infections, medications, and food, where infections were reported to be implicated in 0.5–81% [16,21,22].

AU is considered a manifestation of viral infections especially in children but also in adults [23,24,25,26,27,28]. Viral upper respiratory and gastrointestinal infections were reported as the most frequent cause of AU in 37% of adults [29] and 57% of children [6].

The seasonality of AU supports the role of viral infections in AU, particularly in children. A recent study indicated that respiratory infections were frequently associated with AU and shared a similar seasonality. It also confirmed that the frequency of AU is higher in the autumn (163 children, median age of 4 years) [4,26,30,31].

A systematic review published in 2016 reported a total of 50 cases of urticaria subsequent to viral infection, distributed equally for sex and age, and thus concluded that viral infections can be a potential trigger and/or the main etiologic driver in AU or CU; this was supported by the resolution of urticaria after either healing or control of the viral infection [32]. Kulthanan et al. reported that more than 90% of AU cases had complete remissions within 3 weeks, with the longest duration (median = 8.5 days) when AU was considered idiopathic while AU associated with infections had a median duration of 5 days [5]. The interval between the start of the symptoms of infection and urticaria onset was approximately 6 days for most of the patients (~90%). In total, 1 in 10 patients had urticaria before the onset of symptoms of infection, as was reported in some COVID-19-induced AU cases [33]. Overall, infections subsided about 2 to 3 days before the remission of the urticaria symptoms [5]. In 1999, a study showed that AU during an infectious disease was significantly associated with a high titer to enterovirus by complement fixation and that urticaria was associated with angioedema in 38.6% of the cases [24]. 

Angioedema frequency in urticaria after an infection has not been reported in detail; however, isolated cases of angioedema were reported after Epstein–Barr virus (EBV) infection and COVID-19 [34,35].

A Chinese study evaluating 72 infection-associated AU cases reported significantly elevated white blood cells (WBC) and neutrophil counts, erythrocyte sedimentation rate (ESR), serum C-reactive protein (CRP), and procalcitonin [36].

### 2.2. Which Viral Infections Can Cause Acute Urticaria?

Viral upper airway infections, rather than other viral infections, are most commonly linked to AU, probably because these are the most common infections in humans (Table 1). However, no studies have investigated this point in detail. The most common viruses that cause upper airway infections included rhinovirus, adenovirus, coronavirus, and influenza virus. In most if not all cases of upper airway viral infections, the exact type of the causing virus is usually not identified. The same holds true for gastrointestinal infections, for example, enterovirus.

In a report analyzing 57 infantile/early childhood AU cases, 18 (31.6%) cases were linked to viral infections with the following frequency; adenovirus (5), EBV (5), enterovirus (3), respiratory syncytial virus (RSV) (3), rotavirus (1), and varicella zoster virus (1) [37].

In adults, hepatitis viruses were the most common likely cause of AU, whereas infections due to herpesviridae and respiratory viruses were most common in children [32]. AU cases have also been reported with dengue fever [38,39], norovirus [40], parvovirus B19 [41], herpesviridae (most frequently cytomegalovirus (CMV), human herpesvirus 6 (HHV-6), and EBV in children) [32,42]. In most cases, AU was resolved either after treatment of the infection or after its spontaneous resolution. 

Urticaria was reported by patients in 1.4% of the community-acquired COVID-19 in the initial phase of the pandemic [43]. Recently, several reports described AU and or angioedema associated with SARS-CoV-2 infection (COVID-19) [44]. The role of SARS-CoV-2 in AU and CU is discussed in detail later in this article.

Interestingly, AU has also been described as a prodromal manifestation of viral infections, e.g., of hepatitis B [45]; wheals in urticaria associated with hepatitis-induced jaundice may appear yellow which is called “yellow urticaria” [46].

### 2.3. The Role of COVID-19 as a Cause of Acute Urticaria

The clinical course of COVID-19 ranges from asymptomatic or mild upper respiratory tract symptoms to severe respiratory distress and death. Frequent symptoms at onset are fever, a cough, fatigue, anosmia, myalgia, sputum production, a headache, hemoptysis, and diarrhea. More than 80% of all patients with SARS-CoV-2 infection have a mild form of the disease; however, about 15% to 20% of the patients require hospitalization and up to 5% develop a life-threatening pneumonia. COVID-19 pneumonia may lead to respiratory failure and death due to the development of acute respiratory distress syndrome (ARDS) and multiorgan failure [47].

Cutaneous manifestations of COVID-19 occur in 1% to 5.7% of cases [48,49,50] and the five most common patterns of skin lesions identified in patients with COVID-19 were: pseudo-chilblains (40.9%), maculopapular rash (27.9%), urticarial (12.5%), vesicular (10.7%), and vaso-occlusive lesions (4.4%) [48,51].

The urticarial rash associated with COVID-19 was reported in numerous publications ranging from 1.9% to 19% [43,49,50,52,53]. Urticaria usually appears simultaneously with systemic symptoms, lasts ~1 week, and predominantly involves the trunk and the limbs and spares acral sites affecting mostly adolescents/young adults [51,54]. Of note, a urticarial rash can develop before the onset of COVID-19 symptoms and when present, along with pyrexia, can serve as a clue to its diagnosis [52,55]. Likewise, in a systematic review of 899 patients with COVID-19, 83 (7.9%) had urticarial lesions along COVID-19 and 17 (16%) had urticaria before the onset of COVID-19 symptoms [53,56]. In one study, among other cutaneous manifestations, urticaria manifested with the highest D-dimer levels in COVID-19 patients [54,55,57,58]. Additionally, an analysis of 200 COVID-19 patients with cutaneous manifestations found a significant association between urticaria and gastrointestinal symptoms [56,59]. A correlation between urticaria and the severity of COVID-19 has not been established; however, its occurrence appears not to worsen the prognosis [57,60]. Urticaria multiforme and urticarial vasculitis have also been described in association with COVID-19 [58,59,61,62].

A literature review until 24 July 2020 by Algaadi using “urticaria” or “urticarial” lesions and “COVID-19” or “coronavirus diseases” retrieved 30 articles with a total of 202 patients with COVID-19-associated urticaria [60,63]. Patients’ age ranged from 2 months to 84 years. Gender was reported in 149 patients: 96 (64%) females and 53 (36%) males. Angioedema was reported in only two (0.99%) patients. The skin rashes were treated with antihistamines and systemic and topical steroids and responded well to treatment in all reported cases. The level of care was reported in 129 (64%) patients; of these, 14 (11%) needed treatment in the intensive care unit (ICU), 115 (89%) were treated either as outpatients or inpatients with standard care, and 2 (0.99%) patients died. Of 105 patients, 11 (10%) had urticarial eruptions that developed before the onset of classical COVID-19 signs and symptoms (cough, dyspnea, fever, anosmia, and ageusia) and 47 (45%) developed the rash concurrently with the symptoms.

A multicenter study that evaluated the duration of cutaneous manifestations of COVID-19 in long-haulers (=long covid: patients with dermatological signs of COVID-19 that persisted for more than 60 days) revealed that urticarial eruptions lasted a median of 4 days with a maximum duration of 28 days [61,64]. However, the onset of was also described several weeks after COVID-19 symptoms [55,57,58,60].

## 3. The Role of Viral Infections Including COVID-19 in Chronic Urticaria

### 3.1. Which Viral Infections Can Cause Chronic Urticaria?

CSU is a multifactorial disease in which autoimmunity is the main underlying mechanism [4]. Many patients with CSU report an exacerbation of their disease in response to various triggers such as infections, NSAIDs, or stress via skin MC activation. However, these factors are aggravators or disease modifiers rather than the underlying cause of CSU [4]. 

Viral infections, particularly viral hepatitis, HIV, and herpes viruses were discussed as comorbidities and possible causes of CU, especially CSU (Table 1). In contrast to AU, which can occur as one of the first signs of acute viral hepatitis A and B [45,62,65], chronic hepatitis B and C infections are rarely seen in CSU patients (in <5% and <2%, respectively) [63,66]. Similarly, hepatitis C infection accompanied by chronic urticarial rash was shown in ≤3% of patients and antiviral treatment of hepatitis C coincided with CSU improvement only in two out of nine patients. Although in one study a higher risk for hepatitis B and C was shown for urticaria patients versus control subjects [64,67], this study did not differentiate between AU, inducible urticaria, and urticarial vasculitis. Urticarial vasculitis is known to be linked to chronic hepatitis C and improved in many patients upon antiviral therapy associated with a cure or improvement of hepatitis [65,68].

Several publications investigated a link between human immunodeficiency virus (HIV) infection and CSU [66,67,68,69,70,71,72]. However, the risk of CSU was not associated with HIV infection in a large cohort of CSU patients [69,72]. A few cases of recurrent genital herpes simplex infection were linked to exacerbation of CSU. CSU improved after treatment with acyclovir or raltegravir, a retroviral integrase inhibitor. Other viruses associated with CU are norovirus, parvovirus, HHV-4, and HHV-6, though the role and relevance of these infections in CSU patients are still unclear [40,70,71,72,73,74,75,76].

Some attempts have been made to assess the underlying mechanism of viral infections in the development of CU. The reactivation of a latent herpes virus was hypothesized as a possible mechanism that associates viral infections with CSU. Serology data obtained from omalizumab-dependent severe CSU patients were associated with previous HHV-6 infection, persistent viral gene expression, and replication. These patients also exhibited elevated ELISA titers to EBV antigens compared to controls but not all CSU patients were infected with EBV [73,76]. Some authors demonstrated that endogenous, viral, and bacterial products, particularly protein Fv, an endogenous protein synthesized in the human liver and increased during viral hepatitis, can act as a superantigen (superallergen) by binding to IgE of the VH3 family and activating human basophils and MCs. Similarly, the envelope gp120 of HIV-1 was shown to induce the release of proinflammatory mediators and cytokines from MCs [74,77]. However, there is not enough data that explain the exact role and importance of ‘superallergens’ produced by hepatitis C virus and HIV in the development of CSU.

To summarize, the primary role of viral infections in CU from evidence-based data is still not clear and needs to be further investigated. Viral infections are rare comorbidities of CU, unlikely to be a cause of CSU, but they should be excluded in a patient with a long-standing refractory urticarial rash and urticaria vasculitis. On the other hand, some viral infections, for example, COVID-19, can result in the exacerbation of CSU although information from other viruses is lacking in the literature despite frequent observations of this in the clinical setting [75,78].

**Table 1 viruses-15-01585-t001:** Viral infections implicated/associated with acute and chronic urticaria.

Type of Urticaria	Viral Infection
**Acute urticaria**	Parainfluenza [76,79], Herpes virus [42] (HHV-1, HHV-2, HHV-6, EBV, CMV), Coronaviruses including SARS-CoV-2 [43,77,80], Hepatitis A [78,81], B [45], C [79,82], Adenovirus [37], RSV [37], Dengue virus [38], VZV [37], Parvovirus [41], Rotavirus [37], Norovirus [40], Enterovirus [37]
**Chronic spontaneous urticaria**	Hepatitis A [80,83], B [81,84], C [79,82], Herpes viruses (Herpes simplex [70,73], HHV-6 [73,76], Norovirus [40], Parvovirus [72,75]
**Cold urticaria**	HIV [82,85], EBV [83,86], HBV [84,87], CMV [85,88]
**Cholinergic urticaria**	SARS-CoV-2 [86,89]

EBV: Epstein–Barr virus. CMV: Cytomegalo virus RSV: Respiratory syncytial virus VZV: Varicella zoster virus. HIV: Human immunodeficiency virus SARS-CoV-2: Severe acute respiratory syndrome coronavirus 2.

### 3.2. COVID-19 as a Cause and Trigger of Chronic Urticaria

Although reports on COVID-19-induced CU are scarce, a recent case series from five UCARE centers described 14 cases of COVID-19-induced CSU cases which appeared with an average of 18 days following COVID-19 [9]. The mean age was 33 and 64.3% of patients were females while the mean duration of disease was 16 months. Ten (71.4%) patients had stand-alone CSU and four had accompanying inducible urticaria (2 symptomatic dermographism, 1 cholinergic urticaria, and 1 delayed pressure urticaria plus symptomatic dermographism), where four (28.6%) patients had both wheals and angioedema and no patients had standalone angioedema. CSU was controlled with antihistamines in 78.6% of the patients and three (21.4%) patients required omalizumab treatment (300 mg/month). The authors discussed the possible autoimmune mechanisms underlying the progression of AU to CSU with the high rates of autoimmune markers in their cases; however, there was no control group to test this hypothesis [9]. Additionally, a case with cholinergic urticaria was reported to appear one week after the resolution of COVID-19 symptoms and responded to treatment with antihistamines [86,89].

The effects of COVID-19 on CU patients were evaluated by the UCARE network in a multicenter study [10]. Of 79 CU patients who had COVID-19, 76% were female, with a mean age of 43.0 ± 12.1 years and the most common comorbidity was hypertension (17%). Of these patients, 96% showed a mild course of COVID-19, 14% received inpatient care, and 86% received outpatient care. Only three patients had severe disease and none died. The course of CU did not change in 56% of the patients, while 37% experienced exacerbation and 7% improved. Patients with COVID-19-induced exacerbation of their CU more often required hospitalization due to the course of COVID-19. Conversely, the rate of CU exacerbation in hospitalized patients was higher than in non-hospitalized patients (73% vs. 31%). CU exacerbations have also been reported in a Romanian cohort of CSU patients in whom 44% of infected patients had increased severity of the disease, more frequently seen in moderate–severe COVID infection—47% [75,78]; likewise, high fever was associated with the exacerbation of CU in a Brazilian cohort [87,90]. In the UCARE study, treatments at the time of COVID-19 diagnosis were antihistamines (55.7%), omalizumab (33%), and cyclosporine (2.5%). There was no statistically significant relationship between CU treatment and COVID-19 severity (mild vs. severe disease). No change in CU treatment was undertaken in 71% of the patients following COVID-19 diagnosis which was in accordance with the suggestions of the European Academy of Allergy and Clinical Immunology (EAACI) that discontinuation of omalizumab treatment is not necessary for mild-to-moderate COVID-19 courses [88,91]. Similarly, other studies reported the safe use of omalizumab during the pandemic in CU patients [89,90,92,93] while others highlighted the increased and safe home use of omalizumab [91,94]. Some authors raised the hypothesis that omalizumab use may decrease susceptibility to SARS-CoV-2 given the anti-viral effects of omalizumab that work by decreasing plasmacytoid dendritic cell (pDC) FcεRIα protein expression, thereby increasing IFN-α responses to viral infections such as rhinovirus and influenza infections [92,95].

The COVID-19 pandemic, especially during the times of lockdown and pre-vaccination, had a significant impact on the mental health of CSU patients. Fear of getting COVID-19, anxiety, depression, and stress increased urticaria activity in patients with mild-to-moderate CSU even though they were not infected, as shown by Beyaz S et al. [93,96]. These findings highlight the importance of psychological support for patients with CSU during the pandemic to control disease activity.

## 4. Vaccines and Urticaria

### 4.1. Vaccines as a Cause of AU

Urticaria is a common vaccine adverse effect often monitored in vaccine safety studies and can occur within minutes after vaccination. However, diagnostic coding data post-vaccine vary in reliability; thus, precise rates of AU after vaccination are poorly defined [94,95,97,98]. Acute urticaria after flu vaccination was seen in 8424 out of 152,627 (5.52%) in the Vaccine Adverse Event Reporting System (VAERS) [99]. Delayed urticaria and/or angioedema, as well as nonspecific skin rashes, were reported in 5% to 13% of patients receiving vaccines containing toxoids [96,97,100,101]. Tan and Grattan reported that vaccines were the third most common cause of drug-induced urticaria. From 1238 reports, the three most frequent vaccines associated with urticaria were group C meningococcus conjugate, hepatitis B, and Haemophilus influenzae B [98,102]. The main information on vaccine-induced AU comes from the recent COVID-19 pandemic, therefore, we will focus mainly on COVID-19 vaccines induced AU.

### 4.2. COVID-19 Vaccination as a Cause of AU

The spectrum of cutaneous adverse reactions (CAR) post-COVID-19 vaccines are broad and the pooled incidence of CARs is 5%, ranging from <0.01% to 19%; however, they are less common when compared to local skin reactions (i.e., pain, redness, and swelling at the vaccination site) and systemic adverse events (i.e., fever, fatigue, headache, chill, vomiting, diarrhea, nausea, and arthralgia) [99,103]. A recent meta-analysis reported that the most common CAR after COVID-19 vaccination was an acute injection site reaction (72.2%), rash/unspecified skin eruption (13.8%), urticaria/angioedema (6.5%), pruritus (2.3%), delayed large local reactions (1.9%), maculopapular rash (0.5%), herpes zoster (0.4%), oral blister/ulcer (0.36%), pityriasis rosea/pityriasis rosea-like lesions (0.24%), vesiculobullous lesions (0.2%), petechia/purpura/ecchymosis (0.14%), chilblains/chilblains-like lesions (0.13%), and vasculitis/vasculitis-like lesion (0.1%). mRNA-based COVID-19 vaccines were found to have a higher prevalence of these cutaneous reactions than any other vaccine types at a rate of 6.9% [100,104].

AU, both localized at the injection site or generalized with and without angioedema, is one of the most common cutaneous adverse events reported after COVID-19 vaccination [100,104]. AU was reported after the application of all the main COVID-19 vaccine groups, including whole inactivated virus, adeno-vectored, and most commonly, mRNA-based [100,104]. The reported frequencies with various vaccines were mRNA vaccine at 6.55%, viral vector vaccine at 6.09%, and inactivated viral vaccine at 14.84%.

Recently, from 414 cutaneous reactions reported after COVID-19 vaccination, delayed large local reaction (60.1%), local injection-site reaction (54.2%), and urticaria (6.7%) were most common after the Moderna vaccine, while local injection-site reaction (22.5%) and urticaria (19.7%) were most common after the Pfizer vaccine [101,105]. AU occurred in 23 cases after the Moderna vaccine (18 occurred > 24 h after vaccination) and in 17 cases after the Pfizer-BioNTech vaccine (16 occurred > 24 h after vaccination); none of the urticarial reactions were classified as an immediate hypersensitivity rash. Of the 18 patients who had urticaria after their first vaccine dose, 4 experienced recurrent urticaria with their second dose [101,105]. A more recent analysis from the Vaccine Adverse Event Reporting System (VAERS) database reported urticaria as the second most frequent cutaneous reaction occurring in 32% of the cases. The patients with a history of urticaria or a previous urticarial reaction to a vaccine were both two times more likely to report urticarial eruptions in response to the vaccine in this analysis [102,106].

Regarding the urticaria and angioedema cases reported in the aforementioned global analysis, the average age was 40.7 years, with a 10/36 male/female ratio, onset on an average of 6.5 days after vaccination, and a duration of 24.3 days [100,104]. Another analysis reported that a history of urticaria was present in 18.6% of patients with AU reactions to the COVID-19 vaccine [103,107]. In a report from the COVID-19 Vaccine Allergy Case Registry (allergyresearch.massgeneral.org), urticaria/angioedema cases consisted of 12% of all reactions and were predominantly observed in females (83%) and individuals of the white race [96,99]. Most cases entailed onset after 4 h and were mild and short lasting [104,108].

Angioedema occurs much less often, is usually accompanied by wheals, and is least frequent as an isolated symptom (1.3%) [102,106].

In an analysis of 12 patients who had a delayed systemic urticarial eruption (>4 h after vaccination) after a dose of an mRNA COVID-19 vaccine, the median time to symptom resolution was 4 days. In addition, 9 patients showed negative results to COVID-19 vaccine excipient skin testing and 4 out of 10 patients who received their next mRNA COVID-19 vaccine dose experienced recurrent delayed urticaria [105,109].

From an analysis of 251 allergic reactions reported in South Africa after the Janssen-Ad26.COV2.S vaccine, isolated urticaria and/or angioedema were reported in 70 cases with a median onset of 48 h postvaccination that necessitated treatment with antihistamine (63%) and/or systemic/topical corticosteroids (16%) [106,110].

The onset of AU post-vaccination can be immediate (<6 h), alone or in the context of anaphylaxis, or delayed in onset (>24 h) [106,107,110,111]. However, as previously mentioned, many of the urticaria and angioedema cases recruited to the clinical studies occur >4 h after vaccination and are considered to occur because of non-IgE-mediated mechanisms and a host’s immune inflammatory response to a vaccine rather than being the result of an immunoglobulin (Ig) E-mediated allergy or other hypersensitivity reaction to the vaccine or its excipients. In contrast, as defined by the European Network for Drug Allergy and the European Academy of Allergy and Clinical Immunology, immediate reactions to the COVID-19 vaccines mostly occur within an hour following the injection when the major excipients that may cause severe allergic reactions to COVID-19 vaccines have been reported, i.e., polyethylene glycol (PEG), polysorbate 80, and tromethamine [108,112].

A study from Thailand which presented seven patients who developed urticaria within 4 h after the first vaccination with CoronaVac reported successful revaccination of these patients without a graded challenge, although 5/7 had recurrent urticaria but not systemic reactions. Given that the skin tests (skin prick tests and intradermal tests) and basophil activation test (BAT) results were negative in all cases, the authors suggested that the immune response to vaccines may be responsible for urticaria as the rash still developed even after switching to the other type of vaccine in some cases. They concluded that AU alone should not be a contraindication for the second dose of COVID-19 vaccine [109,113].

AU after COVID-19 vaccination most often occurs after the 1st dose and tends to not recur with subsequent doses. A study including over 49,000 healthcare workers who received mRNA vaccines reported that from 0.4% who developed urticaria after the 1st dose, only 3.3% had a recurrence of urticaria [110,114]. Another study reported that allergic symptoms after the 1st dose of mRNA vaccines may contribute to incomplete vaccination, although out of 1261 with self-reported allergic symptoms after the 1st dose, only 17% had recurrent allergic symptoms at the 2nd dose which were non-severe [111,115]. Current recommendations suggest that delayed rashes and cutaneous reactions that occur > 4 h after vaccination are not a contraindication to subsequent vaccination [108,112].

A recent study invalidated the concerns and hesitations of urticaria patients on COVID-19 vaccines’ adverse effects. Cohen et al. compared the constitutional and dermatologic adverse effects of the COVID-19 vaccines versus the hepatitis B virus (HBV) and seasonal influenza (Flu) vaccines on VAERS, a national, self-reported surveillance database [99,112]. They found that Moderna (3.9%) and Pfizer (3.9%) vaccines have a lower incidence of urticaria compared to the flu (5.5%), and HBV (6.4%) vaccines.

### 4.3. The Impact of Vaccination on CU

Although vaccines have been implicated as a common cause of AU, reports and studies that have investigated vaccines as a cause or trigger of CU are sparse.

Magen et al. studied 1197 patients with CSU and identified 14 patients, i.e., less than one percent, who developed symptoms within 12 weeks after being vaccinated for HBV (3; 21.4%), influenza (4; 28.6%), yellow fever (2; 14.3%) and diphtheria, tetanus, and pertussis (DTP) (2; 14.3%) suggesting an association between these events. A positive ASST (64.3%), anti-nuclear antibodies (42.8%), and thyroid autoantibodies (71.4%) as well as symptoms like myalgia, arthralgia, chronic fatigue, orthostatic intolerance, and sleep disorders were also observed in most patients, indicating that CSU could be part of an autoimmune/inflammatory syndrome induced by adjuvants (Shoenfeld or ASIA) [113,116]. However, the data presented are limited and not sufficient to confirm this cause–effect relationship.

A multicenter study retrospectively compared reports of adverse events following yellow fever (YF) vaccine, an attenuated virus vaccine, in 48 CU patients (30 under omalizumab treatment) and 30 healthy controls. Most patients (89.6%) reported no adverse events in response to the vaccine. Local reactions, asthenia, and angioedema were reported by 6.2%, 2.1%, and 2.1% of the patients, respectively, but no differences were observed when comparing CU patients and healthy controls. The safety of the YF vaccine in patients under treatment with omalizumab was also analyzed and those patients did not have a higher rate of adverse events, including CU exacerbations when compared to CU patients treated with antihistamines. In both groups CU was controlled, suggesting that the YF vaccine is safe for CU patients and CU patients treated with omalizumab [114,117].

### 4.4. COVID-19 Vaccination as a Cause of CU

Along with the short-lasting AU cases triggered by COVID-19 vaccination, there has been an increasing emergence of CU and angioedema cases following COVID-19 vaccination [115,116,117,118,119,120,121]. Those reports included mostly women, ages ranging between 20 to 78 with symptoms occurring from 4 days to 3 months after vaccination. In some of these, a case of symptomatic dermographism was reported and some but not all cases had accompanying angioedema [119,120,122,123].

Magen et al. analyzed 32 patients with new-onset persistent urticaria and 27 CSU patients in remission that relapsed within three months after BNT 162b2 mRNA vaccination. They found that a positive autologous serum skin test and basopenia in the peripheral blood were positively associated with the likelihood of CSU recurrence after vaccination with BNT162b2 mRNA. Furthermore, the relapsed CSU and new-onset CSU groups had more allergic comorbidities. They concluded that it is possible that BNT162b2 mRNA vaccination serves as a provoking and/or relapsing factor of CSU in individuals with allergic diseases and/or predisposed autoimmunity [115,118].

In a case series of 32 patients who developed CU after repeated immunization with SARS-CoV-2 mRNA vaccines, Pescosolido et al. [121,124] reported that symptoms started after the third dose of vaccine in 94% of patients; after a symptom-free interval of at least 48 h, urticaria developed at a median of 10 days after immunization. Most of the patients (78%) responded to a single daily dose regimen of antihistamines. Authors could not identify basophil activation by patient’s serum factors; however, BAT assessed in seven patients was positive for both tested mRNA vaccines and more than half of patients showed a marked basophil activation with polysorbate-80 which is commonly used as the solubilizer or emulsifier. These findings pointed out that patients might have become sensitized to vaccine compounds after repeated immunization (type 1 hypersensitivity) or that the antigens and antibodies triggered by the vaccine cause urticarial lesions in the context of immune complexes activating basophils on a C3a and C5a driven mechanism due to hyperimmunization (type 3 hypersensitivity).

Duperrex et al. [122,125] analyzed 862 cases of post-SARS-CoV-2 vaccine CSU cases which occurred mainly after the 3rd dose (booster) of mRNA vaccines. The overall crude incidence rate of CSU after a SARS-CoV-2 booster per 100,000 persons immunized with a booster was 24 and 19 in the two cohorts, respectively. Compared with the Pfizer-BioNTech vaccine, the relative risk of developing CSU after the Moderna vaccine was a mean of 18.5 in the two cohorts. However, the selection bias for patients with CSU in relation to SARS-CoV-2 vaccines and the confounding effect of the Omicron variant that was prevalent at the time the study was conducted (31% verified COVID-19) were the limitations of the study.

A recent study from Korea analyzed patients who developed allergic reactions after SARS-CoV-2 vaccination [123,126]. A total of 129 patients [42 had an immediate allergic reaction (≤24 h) while 87 developed delayed urticaria (>24 h)] were compared to a control group who did not develop any cutaneous reaction after 8 weeks of at least two doses of SARS-CoV-2 vaccination (*n* = 115). With an ex vivo BAT and LAT assay, the authors showed that the spike protein and the excipient of mRNA-1273 vaccine-tromethamine are the culprit allergen(s) for patients with immediate allergic and urticarial reactions induced by SARS-CoV-2 vaccines. They reported that out of the cohort, 57 patients later developed CU and these patients showed a hyperinflammatory condition with elevated cytokines and chemokines; 81.3% of them (antihistamine-resistant) showed ex vivo BAT positivity when stimulated with patients’ autoserum. They identified higher rates of underlying thyroid disease and upregulated levels of total IgE, IgE-anti-IL-24, IgG-anti-TPO (but not IgE-anti-TPO), IgG-anti-FcεRI, IgG-anti-Thymidylate Synthase (TYMS), and IgG-anti-Thyroid hormone receptor alpha (THRA) autoantibodies in patients with CU induced by SARS-CoV-2 vaccines, suggesting that CU after SARS-CoV-2 vaccination could be due to autoreactive/autoimmune reactions triggered by SARS-CoV-2 vaccines. They also showed that these CU patients triggered by SARS-CoV-2 vaccines respond favorably to treatment with omalizumab. In their cohort of 129 patients who had experienced SARS-CoV-2 vaccines-induced immediate allergy and CU, approximately 6.3–20% of patients further relapsed itchy and skin rash after following re-vaccination and they suggested that performing ex vivo tests to identify the main culprit component or excipient of vaccines may help to decrease the incidence of cutaneous reactions induced by SARS-CoV-2 vaccines [123,126].

### 4.5. The Impact of SARS-CoV-2 Vaccination on CU

Another concern with the SARS-CoV-2 vaccination has been the anxiety of CU patients as to whether vaccination would exacerbate their urticaria or would cause systemic allergic reactions. In an attempt to answer these questions, a few small-sized studies have been carried out. Grieco T et al. identified worsening of urticaria in 8% of their CU patients with a mean symptom duration of 2 days and 11 h, mostly after the first dose and mainly managed by antihistamines. The authors also observed that fewer exacerbations happened when patients were under omalizumab therapy [124,127]. Likewise, Lascialfari et al. [125,128] reported that relapse or exacerbation of urticaria after SARS-CoV-2 vaccination was observed in only 7.7% of pediatric CSU patients which lasted for less than one week. Picone et al. [126,129] reported postvaccine urticaria to flare up in 8% of CSU patients who were well controlled under antihistamine and omalizumab treatment, while Tuchinda et al. [127,130] reported a higher rate (15%). Others found no effect of mRNA-SARS-CoV-2 vaccination on urticaria activity scores in 28 patients with CSU [128,131].

The largest study that aimed to determine urticaria exacerbations and rates of systemic reactions in CU patients was carried out by the UCARE network [12]. It included 2769 CU (CSU 70.9%, CIndU 12.2%, CSU + CIndU 16.9%) patients who were vaccinated with a minimum of one dose of SARS-CoV-2 vaccine from 50 UCARE centers in 26 countries. Most of the patients (90%) received at least two SARS-CoV-2 vaccine doses and most were vaccinated with the Pfizer-BioNTech (BNT162b2) vaccine followed by Oxford/AstraZeneca. CU exacerbation was reported by 9% of the patients [456 patients; a total of 527 reactions (527/5877]]; specifically, in 8% (223/2769) of patients after the first dose, in 9.6% (234/2445) after the second dose, in 11% (70/637) after the third dose, and none of the 26 patients after the fourth dose. Although the risk of CU exacerbation was found to be increased in patients who had CU exacerbation in the previous vaccine administration, half of the patients who had CU exacerbation after the first dose and 70% of the patients with CU exacerbation after the second dose did not have CU exacerbation at the following doses, respectively. Fifty-five patients (9.8%) did not experience CU exacerbation at the first and second doses but had CU exacerbation only at the third dose. Allergic reactions and CU exacerbation were rare despite the low rate of premedication before vaccination (~5%). CU exacerbation most commonly occurred within 48 h (59.2%) following vaccination and tended to last for a maximum of a few weeks in almost 70% of patients while others lasted a few weeks to a few months. CU exacerbations were mainly treated with antihistamines (70.3%) while only 10% required systemic glucocorticosteroids.

The risk factors were female gender, disease duration shorter than 24 months, having CSU vs. CIndU, receipt of adenovirus viral vector vaccine, NSAID/aspirin intolerance, and having concern about getting vaccinated; being under omalizumab treatment during vaccination and having Latino/Hispanic ethnicity decreased the risk of CU exacerbation. Interestingly, CU exacerbation was found to be higher in patients who had constitutional adverse effects following vaccination. The authors proposed that CU exacerbation occurs as the host’s immune and inflammatory response to SARS-CoV-2 vaccines rather than a hypersensitivity reaction since no features or autoimmunity (such as the presence of autoimmune thyroid disorders, anti-TPO positivity, low IgE, and eosinopenia) nor atopy (presence of atopic disorders, high IgE, and prick test positivity) were associated with CU exacerbation in this study. Severe allergic reactions were reported by seven (0.25%) patients although the relevance of these to anaphylaxis remained to be determined and risk factors included previous anaphylaxis and other allergies such as drug, dust mite, pollen, etc. [12].

Type of urticaria reported to be associated with COVID-19 infection and SARS-CoV-2 vaccination are presented in Table 2. 

## 5. Possible Pathogenic Mechanisms of Virus and Mast Cell Interactions Involved in the Occurrence of AU and CU Urticaria Exacerbations with COVID-19 and SARS-CoV-2 Vaccinations

### 5.1. Potential Mechanisms of Mast Cell Activation by Viruses

Even though urticaria was reported as one of the most frequent cutaneous manifestations observed in patients with SARS-CoV-2 infection [60,63], the molecular and cellular mechanisms underlying the development of urticarial symptoms in COVID-19 patients remain poorly understood.

Due to their location at the interface between the host and sites of pathogen entry, MCs are one of the first cells of the immune system that can recognize and rapidly respond to viruses such as SARS-CoV-2 [14]. MCs can not only directly sense viruses through the expression of TLR3, TLR7, TLR8, TLR9, melanoma differentiation-associated protein 5 (MDA5), and retinoic acid-inducible gene (RIG)-I [14,129] but can also be activated by tissue damage and stress through alarmin and purinergic receptors upon infection [14,130,131,132,133,134]. Upon activation by viruses, MCs release some specific chemokines, such as the ligand 5 (CCL5), which leads to virus-specific CD8+ T cells’ activation and production of antiviral cytokines [132,133,135,136]. Additionally, RNA viruses stimulate MCs to produce IFN-γ and CXCL8 chemokines, resulting in the recruitment of NK cells that also produce type I-IFNs which are anti-viral cytokines [134,137]. Besides the positive effects of MCs in fighting infections, on the other hand, viruses stimulate mucosa MCs to release pro-inflammatory cytokines such as IL-1, TNF, IL-6, and proteases which aggravate the inflammatory state. Therefore, viral infections might lead to dual effects on MCs: a positive effect by helping the immune system to fight infection and a negative effect by aggravating the pathological state of the patient via the release of several cytokines, chemokines, serin proteases, lipid mediators, and growth factors [135,136,138,139] that in turn activate other immune cells which is also the case during COVID-19 [137,138,140,141].

Gebremeskel and colleagues [139,142] conducted research to investigate the role of MCs in COVID-19 pathogenesis. Their results showed that MC-specific proteases including β-tryptase, carboxypeptidase-3, and chymase are notably increased in sera collected from patients when compared to that of healthy individuals suggesting a systemic activation of MCs. These authors suggest that rather than direct infection with SARS-CoV-2, it is likely that secondary activation of innate immune cells like MCs and eosinophils via pattern-recognition receptors (PRRs) contributes to the release of inflammatory cytokines since isolated peripheral blood-derived human MCs display low-level expression of the main entry receptor (ACE-2). Synthetic analogs of ssRNA (R848, which is an agonist of TLR-7 and 8) and dsRNA (poly I:C, which is TLR-3 agonist) activate MCs and induce the production of IL-8, CCL3, and CCL4 as well as chymase and tryptase, suggesting that TLR-3, 7, and 8 are involved in SARS-CoV-2-mediated MC activation. Interestingly, administration of a Siglec-8 mAb reversed the TLR-dependent inflammation [139,142].

Two mechanisms might involve MCs in the development of urticaria in COVID-19 patients. Firstly, the development of an immune hypersensitivity response to SARS-CoV-2 RNA and secondly, an inflammatory cascade driven by a hyperactive immune response with complement activation [140,141,142,143,144,145]. High concentrations of TNF, IL-6, IL-1β, granulocyte-macrophage colony-stimulating factor (GM-CSF), and chemokine (C-C-motif) ligand 2 (CCL2), known to be produced by MCs, are found during the course of infection in COVID-19 patients [15,137,140,143,146]; IL-6 has been shown to increase MC proliferation and induce a more reactive phenotype providing a possible link between elevated IL-6 levels and MC activation [144,147]. Furthermore, colocalization of SARS-CoV-2 glycoproteins and respective complement mediators have been reported in peripheral cutaneous blood vessels [145,148]. Thus, it is possible that proinflammatory cytokines such as IL-6 and chemokines as well as complement activation with subsequent C5a and C3a release during SARS-CoV-2 infections may result in MC degranulation and development of urticaria in COVID-19 patients.

Furthermore, IgE antibodies against structural proteins of the SARS-CoV-2 have been recently assessed. In this study, the most frequent IgE antibody detected in COVID-19 patients was against the full length of the nucleocapsid (NFL) of the virus [146,149]. These anti-NFL IgE antibodies were detected in 100, 96.2, and 66.7% of patients with severe disease, moderate, and mild disease, respectively [138]. Moreover, IgG, IgA, and IgE antibodies to SARS-CoV-2 were higher in patients with severe/moderate COVID-19 than in patients with mild disease [146,149].

Additionally, the release of substance P (SP) from immune cells in COVID-19 is augmented, which in this state might stimulate MRGPRX2 receptors on MCs and lead to their activation [147,150].

Thus, the possible mechanisms involved in the activation of MCs by SARS-CoV-2 could be summarized as follows. Firstly, direct activation of MCs by viral particles via the innate immunity receptors (TLR signaling, i.e., TLR3 detection of dsRNA, sphingosine-1-phosphate (S1P) binding to its receptor S1PR, and RIG-I recognition of uncapped vRNA) or through Fc receptors especially in the early lesions [148,151]. Secondly, alarmins and IL-33 are released as a result of the infection of neighboring cells, such as endothelial cells (infected via SARS-CoV-2/ACE2 interaction) or the release of SP from immune cells [143,146,147,150]. Thirdly, activation by cytokines or complement components that migrate to the skin via blood circulation such as IL-6, IFN-α, IL-1, C3a, and C5a [143,146]. Fourthly, the formation of IgE autoantibodies against viral proteins [146,149]. Lastly, the activation of autoimmunity: the immune response, i.e., induced against viral antigens in SARS-CoV-2 infection, could target host molecules that share sequence homology or structural similarities with viral epitopes. Studies have shown that there are similarities and homology between SARS-CoV-2 proteins and human tissue antigens; antibodies to SARS-CoV-2 can bind some human tissue antigens [149,150,152,153]. Figure 2a presents our current knowledge of MC interactions with SARS-CoV-2 in the development/exacerbation of acute or chronic urticaria in COVID-19 patients and after SARS-CoV-2 vaccines. However, the molecular mechanisms underlying the initiation or exacerbation of urticaria symptoms upon SARS-CoV-2 infection need to be better characterized.

### 5.2. Potential Mechanisms of Mast Cell Activation by Vaccines

Whether the above-mentioned mechanisms can also explain the development of urticaria after receiving the AstraZeneca/Oxford (ChAdOx1), Pfizer/BioNTech (BNT162b2), and Moderna (mRNA-1273) SARS-CoV-2 vaccines needs to be further investigated [117,120,151,152,153,154,155,156]. Urticaria/angioedema occurring in the context of vaccination probably occurs as a result of the host’s immune inflammatory response to a vaccine rather than being the result of an immunoglobulin (Ig) E-mediated allergy or other hypersensitivity reaction to the vaccine or its excipients, i.e., polyethylene glycol (PEG), polysorbate, and tromethamine [104,108,112]. Given that in the majority of cases the urticarial reactions described in the literature did not recur with subsequent vaccinations, the skin tests performed with vaccine ingredients were negative [105,109,113]; however, there are also opposing findings which were reported in a recent study by Wang et al. where they found positivity by ex vivo tests with PEG and tromethamine. Furthermore, this was the case for spike protein for the first time, though they did not find a difference between allergic and control patients in terms of anti-PEG IgE and anti-spike IgE antibodies [123,126].

Wang et al. also suggested that the vaccine components activate T cells and result in delayed hypersensitivity or urticarial reactions since they found that many cytokines and chemokines such as IL-2, IL-4, IL-6, IL-17 A TARC/CCL17, PARC/CCL18, and MIG/CXCL9 were elevated in patients who reacted to SARS-CoV-2 vaccines [123,126].

To provide an adaptive response, vaccine antigens induce innate immune responses. Firstly, vaccine antigens are recognized as potential pathogens by pathogen-associated molecular patterns (PAMPs) or damage-associated molecular patterns (DAMPs) and PRRs such as TLR-5 on local or circulating immune cells and resident stromal cells which result in the release of the pyrogenic cytokines IL-6, TNF, and prostaglandin-E2 in the bloodstream. A series of innate immune events occur (such as phagocytosis, the release of inflammatory mediators including chemokines and cytokines, activation of the complement, and cellular recruitment) to trigger an antigen-specific acquired immune response which is necessary for protection against disease. These inflammatory events may also lead to the development of signs and symptoms of unwanted effects such as local reactions, fever, fatigue, and headache) [154,157]. With SARS-COV-2 vaccines, the same events occur, i.e., the mRNA and nanoparticles in the BNT162b2 mRNA vaccine can activate the pleiotropic innate immune system [155,158] and the induction of innate immune responses may contribute to CU exacerbations and the relapse of CSU following vaccination.

The stimulation of immune complexes and the induction of autoantibodies after the vaccination could explain the emergence of CSU in vaccinated patients as part of an autoimmune/inflammatory syndrome induced by adjuvants (Shoenfeld or ASIA) as suggested by Magen et al. [113,116]. The spike protein (SP) antigen has some similarities to human proteins and can trigger an autoimmune response after vaccination against SARS-CoV-2 [156,159]. Molecular mimicry is also a possible mechanism that contributes to SARS-CoV-2 vaccine-associated autoimmune pathology [157,160]. Furthermore, nucleic acids in vaccines could activate factor XII resulting in the generation of bradykinin with the subsequent development of angioedema [158,161]. This hypothesis is also supported by the findings of Wang et al. as they found that levels of IgE-anti-IL24, IgG-anti-TPO (but not IgE-anti-TPO), IgG-anti-FcεRI, IgG-anti-TYMS, and IgG-anti-THRA autoantibodies were elevated in patients with CU induced by SARS-CoV-2 vaccines [123,126].

Thus, urticarial rashes of a short duration or exacerbation of CU after vaccination could be related to unspecific activation of the immune system, whereas the occurrence of CSU after SARS-CoV-2 vaccination could result from SARS-CoV-2 IgE or IgG antibody formation. Further studies exploring the existence of autoantibodies will shed light on the pathophysiology of CU occurring after SARS-CoV-2 vaccinations. Here, it is important to note that autoantibody formation after SARS-CoV-2 vaccination was of a low titer and was transient [159,162], in contrast with autoantibody formation following COVID-19, with data showing at least a single autoantibody that was present in approximately 50% of hospitalized patients [160,163].

IL-6 and TNF can contribute to the cytokine storm in COVID-19 patients. The generation of SARS-CoV-2 IgE antibodies and immune complex formation induce complement, factor XII, and MC activation leading to MC degranulation. C5a generation increases MC degranulation and histamine release which results in vasodilatation, extravasation, and subsequent wheals and angioedema formation (Figure 2b).

## 6. Limitations of Reported Studies

Several important questions remain unanswered and need to be addressed by future research. Firstly, all reports on the rate of viral infection-induced AU are based on retrospective studies, i.e., prospective studies are lacking. As many cases of AU after a viral infection are mild and of short duration, retrospective studies may underestimate the rates of AU caused by viral infections. Secondly, NSAIDs and, less often, antibiotics can induce AU. Patients with AU upon viral infections might develop symptoms triggered by the use of these drugs, the viral infection, or both. One can hypothesize that symptoms are caused by the viral infection and not by the drug itself. In clinical practice, most of these patients are labeled as drug allergic without appropriate testing, mainly due to fear of a life-threatening reaction. In most cases, drug hypersensitivity can be confirmed in only a minority of patients. Thirdly, the rate of CU resulting from AU remains ill-characterized and it is currently unclear if the rates of AU becoming chronic are different in AU due to viral infection versus AU due to other causes. Fourthly, AU following a viral infection is more likely to be coincidental rather than causative. The underlying mechanisms of viral infection-induced AU need to be evaluated before we can establish a causal relationship. Interestingly, treatment or spontaneous recovery of the viral infection were reported to be linked to AU remission, although this may, again, be coincidental and explained by the natural course of AU [32].

It is also noteworthy that COVID-19 is primarily a lung infection where skin MCs can be activated as a secondary event. The involved mechanistic pathways, such as the possibility of the remote lung MCs–skin MCs interaction by releasing mediators in the circulation that links them indirectly, can be interesting themes for further research. Additionally, although after the emergence of COVID-19 ACE-2 was known as the main receptor of the virus, later, further receptors were proposed [161]. Investigation on MC expression of these receptors may be helpful to shed light on the possible mechanism of direct interaction (or MC infection) of SARS-CoV-2 and MCs.

Additionally, the emergence of new-onset urticaria after SARS-CoV-2 vaccinations is increasingly being reported. However, it is important to acknowledge their limitations. Firstly, many studies on urticaria following SARS-CoV-2 vaccination lack appropriate control groups for comparison. Without a control group of unvaccinated individuals or individuals receiving a placebo, it becomes challenging to establish a causal relationship between vaccination and the onset of urticaria. Comparative studies with control groups would provide a more robust understanding of the association. Secondly, some studies have small sample sizes, limiting the statistical power and generalizability of their findings. Thirdly, the absence of comprehensive pre-vaccination data on urticaria incidence and prevalence makes it difficult to establish a baseline and determine whether the observed cases of urticaria are higher than expected. Without pre-existing data, it is challenging to differentiate between urticaria occurring due to vaccination and cases that would have occurred naturally in the absence of vaccination. Fourthly, studies may not adequately account for potential confounding factors that could influence the development of CSU. Factors such as pre-existing allergies, concomitant medications, or underlying medical conditions might influence the occurrence and severity of CSU following vaccination. Comprehensive assessment and adjustment for these confounders are necessary to accurately attribute CSU to vaccine exposure. Fifthly, while studies aiming to elucidate the underlying mechanisms of SARS-CoV-2 vaccination-induced urticaria provide valuable insights, there are some discrepancies among these studies. Addressing these limitations is crucial for improving the quality and reliability of research on urticaria appearance after SARS-CoV-2 vaccination. Future studies should strive for larger sample sizes, well-designed control groups, standardized methodologies, long-term follow-up, and collaboration across multiple research centers and should aim to determine immune markers, such as autoantibody levels, cytokine profiles, and immune cell subsets as well as the involvement of mast cells, basophils, and other immune cells and the activation of specific signaling pathways or the release of inflammatory mediators.

The UCARE network has developed a forthcoming study aimed at investigating the distinguishing clinical and laboratory characteristics of urticaria following COVID-19 or SARS-CoV-2 vaccination. This study holds the potential to offer valuable insights into the mechanistic pathways implicated in COVID-19-induced, SARS-CoV-2 vaccine-induced, and non-COVID-induced urticaria.

## 7. Conclusions

The COVID-19 pandemic provided an opportunity to observe many clinical examples of MC–virus interactions as well as the effects of vaccinations on MC-driven diseases. Viral infections can induce urticaria de novo or urticaria exacerbations as can vaccinations; however, precise estimates are not always clear from available data. During the pandemic, we learned that COVID-19 may lead to urticarial eruptions, AU, CSU rarely, and the exacerbation of CU. The exacerbation of CU during COVID-19 was common at 40% even when during mild illness severity; while SARS-CoV-2 vaccination-induced exacerbations of CU occurred in less than one-tenth of patients and omalizumab treatment was preventative. Since in the COVAC-CU study it was found that having anxiety to get vaccinated increased the risk of CU exacerbation, it is important to communicate the potential adverse effects of the vaccine and to properly address the possibility of disease exacerbation as well as the importance of having the disease under control during vaccinations. However, it is also important to stress that, although SARS-CoV-2 vaccines may be linked with urticaria and the exacerbation of CU, it is not recommended to withhold further immunization and this plausible adverse event may be treated, usually requiring minimal intervention (e.g., anti-histamine). There is certainly a great need to understand the exact pathomechanism of these reactions and to unlock the interactions between MCs and viruses as well as consequences of these interactions.

## Figures and Tables

**Figure 1 viruses-15-01585-f001:**
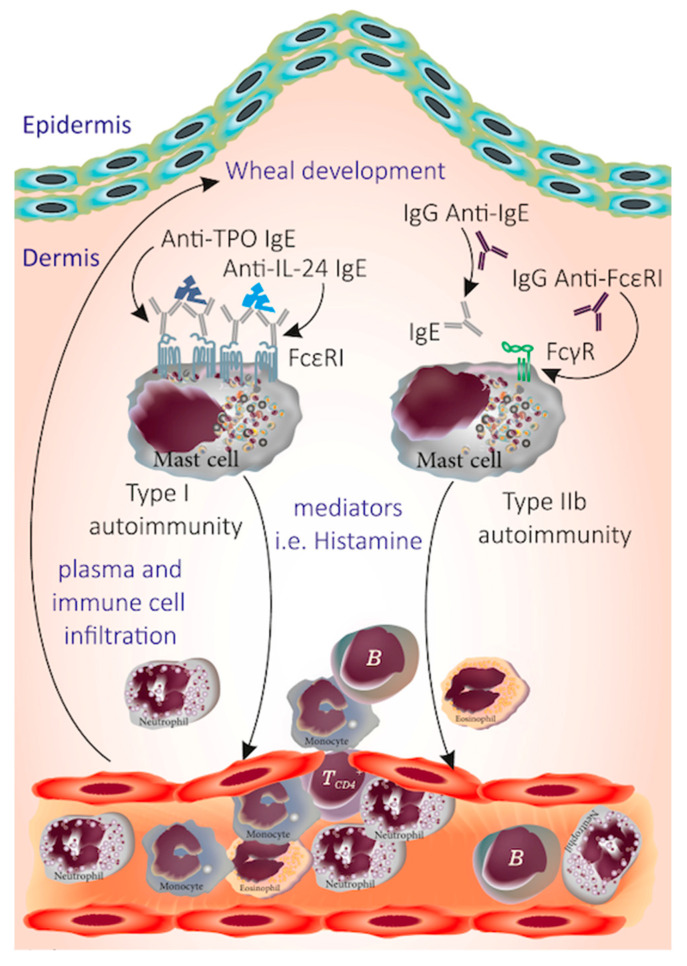
A schematic demonstration of autoimmune mechanisms in CSU. The autoimmunity in CSU can be studied in two main categories of type I and IIb in which either the presence of IgE against autoallergens (TPO, IL-24, etc.) or the IgG autoantibodies against IgE and its receptor FcεRI, respectively, activate dermal MCs and induce the release of mediators such as histamine that mediate cellular infiltration and contribute to the development of wheal.

**Figure 2 viruses-15-01585-f002:**
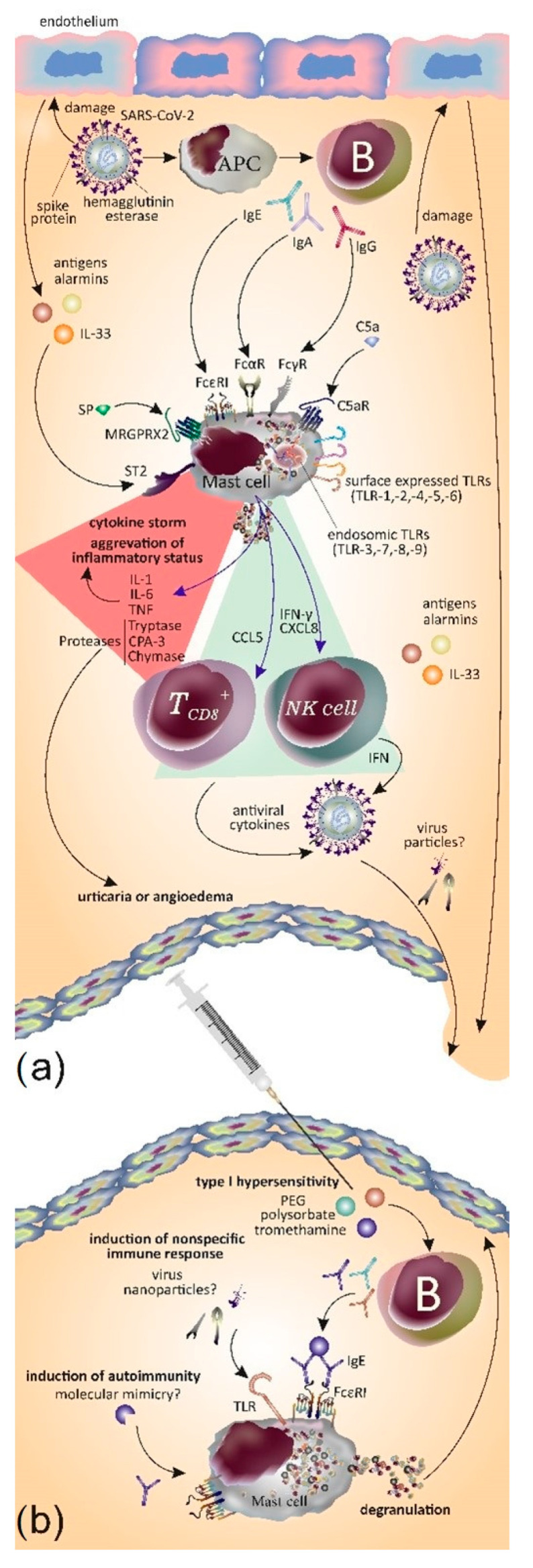
(**a**) MCs play a role in defense against SARS-CoV-2 by releasing CCL5 which activates CD8+ T cells to release antiviral cytokines; also, MCs release IFN-γ and CXCL8 which acts on NK cells and induces the production of IFN. However, the activation of MCs may also lead to detrimental effects. Main mechanisms on how MCs are activated by SARS-CoV-2 are depicted in the figure. SARS-CoV-2 induces endothelium damage and in turn endothelial cells release antigens, alarmins, and cytokines including IL-33 which activate MCs by engaging ST2. Moreover, MCs express TLRs that sense the pathogen antigens. B cells become activated by antigen presenting cells and in turn produce different classes of immunoglobulins such as IgE, IgA, and IgG which binds to MCs. The activation of MCs induces the release of cytokines and proteases that result in a cytokine storm, the aggravation of inflammatory status, and the induction of urticaria or angioedema (red triangle). (**b**) Main mechanisms as to how MCs may become activated upon administration of SARS-CoV-2 vaccines and lead to the formation of urticaria and/or angioedema. 1. Type 1 hypersensitivity: the production of IgE against vaccine components, namely mainly PEG, polysorbate, and tromethamine, activates MCs via FcεRI. 2. Induction of non-specific immune responses: MCs may sense the presence of virus nanoparticles via its expressed PRRs, mainly TLRs, which results in the orchestration of nonspecific immune responses. 3. Induction of autoimmunity: Additionally, mechanisms such as molecular mimicry may induce autoimmunity and the production of IgE capable of activation of MCs.

**Table 2 viruses-15-01585-t002:** Type of urticaria associated with COVID-19 and SARS-CoV-2 vaccination.

Type of Urticaria	COVID-19	SARS-CoV-2 Vaccination
**Acute urticaria**	In total, 1.9–19% of dermatologic reactions Appears mainly simultaneously with the infection Lasts approximately one week Responds well to antihistamines	In total, 6.5–12% of cutaneous adverse reactions; Mainly short lived, i.e., one week More common with inactivated viral vaccines; Tends to not recur with subsequent vaccinations; Most of them occur >4 h after vaccination; Regarded as host’s immune inflammatory response to a vaccine rather than being a hypersensitivity reaction to vaccine ingredients
**Chronic spontaneous urticaria**	Case series of CSU Appears mostly two weeks after infection; Responds well to antihistamines	Many case reports /series; Mainly with mRNA vaccines; Reported to appear after four days to up to three months after vaccination; Responds well to antihistamines; Autoimmune mechanisms are speculated
**Chronic inducible urticaria**	Case report of cholinergic urticaria	Case report of symptomatic dermographism
**Chronic urticaria exacerbation**	Reported in 37–44% of CU cases Increases with severity of COVID-19 Omalizumab use might be preventive	Seen in 8–15% of CU cases; Seems to be more common with viral vector vaccines; Responds well to antihistamines; Omalizumab treatment associated with lower risk of exacerbations

## Data Availability

Data available in a publicly accessible repository.

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
