# Peer review of "How Infection and Vaccination Are Linked to Acute and Chronic Urticaria: A Special Focus on COVID-19"

_viruses, 2023, doi:10.3390/v15071585_

Round 1
Reviewer 1 Report
The review is timely and of interest to the community.
A few overall minor remarks:
1. Page 3, 95. This study showed that COVID-19 vaccinations are safe 95 in CU patients, and only 16% and 0.25% of patients experience an exacerbation of their 96 urticaria and systemic allergic reactions, respectively. 16% is not null. Please remove the “only” to remain neutral.
2. Could the authors comment (briefly) of the role of other infection (even if not viral) such as helminths and H pylori on triggering AU and CU?
3. Cutaneous manifestations of COVID-19 occur in up to 20% of cases This is an overstatement since others report less than 1 percent of cutaneous manifestation, probably strain-dependent. This topic is discussed on uptodate, please check. I think the authors should discuss the cutaneous manifestation of COVID before/after the omicron wave.
4. Table 2. If the authors report the number of cases of CSU related to COVID-19 infection, they should be doing the same with COVID-19 vaccination. 19 cases of CSU after COVID are strikingly low considering the number of infected patients.
5. Could the authors briefly comment on the rate of AU/CU after flu vaccination, considering that 100 million doses are injected every year.
6. Line 460 page 10 Compared with the Pfizer-BioNTech vac- 460 cine, the relative risk of developing CSU after the Moderna vaccine was a mean of 18.5%. Remove %
Author Response
Point-by-point response to Editors and Reviewers
Date: 10-07-2023
Manuscript Number: viruses-2482563
Title of Article: How infection and vaccination are linked to acute and chronic urticaria: with a special focus on COVID-19
Name of the Corresponding Author: Marcus Maurer
Email Address of the Corresponding Author: marcus.maurer@charite.de
Dear Louies,
We thank you for your review and the opportunity to revise our manuscript “How infection and vaccination are linked to acute and chronic urticaria: with a special focus on COVID-19” (viruses-2482563). We diligently made all of the revisions requested by the reviewers as described in our detailed point-by-point response below and highlighted changes in the document keeping the track changes on. All authors agree with the revised manuscript.
We hope that our revisions are regarded as sufficient so that a re-evaluation of the manuscript may now support its publication in Viruses Journal. Again, we thank you and all reviewers for the positive feedback and suggestions, which helped us to improve our manuscript.
With our best wishes,
Emek and Marcus
Main changes done:
- Changes according to reviewers’ suggestions are carried out
- Changes in Table 2 are done, the table has more detailed data now
- References are revised and 2 references are removed
- Limitations part revised and extended
Reviewer 1
The review is timely and of interest to the community.
Reply: We thank the reviewer for their positive evaluation of our manuscript.
A few overall minor remarks:
- Page 3, 95. This study showed that COVID-19 vaccinations are safe 95 in CU patients, and only 16% and 0.25% of patients experience an exacerbation of their 96 urticaria and systemic allergic reactions, respectively. 16% is not null. Please remove the “only” to remain neutral.
Reply: We thank the reviewer for their valuable suggestion, we agree with them. The 16% was mistakenly written: the correct percentage is 9%. Now we changed the percentage and also deleted ‘’only’’. Now the text reads as below:
This study showed that COVID-19 vaccinations are safe in CU patients, and 9% and 0.25% of patients experience an exacerbation of their urticaria and systemic allergic reactions, respectively (12).
- Could the authors comment (briefly) of the role of other infection (even if not viral) such as helminths and H pylori on triggering AU and CU?
Reply: We thank the reviewer for their suggestion. We inserted a sentence about this topic. Now the text reads as below:
The role of H. pylori infection, bacterial infections and bowel parasites in urticaria is not clear however the urticaria treatment guideline suggests the elimination of them if found (1).
- Cutaneous manifestations of COVID-19 occur in up to 20% of cases This is an overstatement since others report less than 1 percent of cutaneous manifestation, probably strain-dependent. This topic is discussed on uptodate, please check. I think the authors should discuss the cutaneous manifestation of COVID before/after the omicron wave.
Reply: We thank the reviewer for their attention and valuable suggestion. We included results from 3 main metaanalysis on the topic which report the rates from 1% to 5.7%. Now the text reads as below:
Cutaneous manifestations of COVID-19 occur in 1% to 5.7% of cases and the five most common patterns of skin lesions identified in patients with COVID-19 were: pseudo-chilblains (40.9%), maculopapular rash (27.9%), urticarial (12.5%), vesicular (10.7%), and vaso-occlusive lesions (4.4%) (48).
- Table 2. If the authors report the number of cases of CSU related to COVID-19 infection, they should be doing the same with COVID-19 vaccination. 19 cases of CSU after COVID are strikingly low considering the number of infected patients.
Reply: Thank you for the suggestion. We removed the numbers and revised as ‘’Case series of CSU’’ and ‘’Case report of cholinergic urticaria’’
- Could the authors briefly comment on the rate of AU/CU after flu vaccination, considering that 100 million doses are injected every year.
Reply: We thank the reviewer for their suggestion. Acute urticaria after Flu vaccination was seen in 8424 out of 152,627 (5.52%) in the Vaccine Adverse Event Reporting System (VAERS). However, there is not enough data on the occurrence of CU after flu vaccine. There is only a study by Magen E that describes association with different kinds of vaccinations and occurrence of CU. Out of 14 patients, 4 (28.6 %) received Flu vaccine. The text is revised as below:
Acute urticaria after Flu vaccination was seen in 8424 out of 152,627 (5.52%) in the Vaccine Adverse Event Reporting System (VAERS).
Magen et al. studied 1197 patients with CSU and identified 14 patients, i.e., less than one percent, who developed symptoms within 12 weeks after being vaccinated for HBV (3; 21.4%), influenza (4; 28.6%), yellow fever (2;14.3 %) and diphtheria, tetanus, and pertussis (DTP) (2;14.3 %) suggesting an association between these events.
- Line 460 page 10 Compared with the Pfizer-BioNTech vaccine, the relative risk of developing CSU after the Moderna vaccine was a mean of 18.5%. Remove %
Reply: Thank you for the suggestion. We deleted the %.

Reviewer 2 Report
This is a well written comprehensive review of the current knowledge regarding plausible associations between COVID-19 infection and/or vaccination and acute/chronic urticaria.
2 minor comments
A. It may be of important to emphasize that although anti-COVID vaccine may be linked with urticaria and particularly exacerbation of CSU - It is not recommended to withhold further immunization and this plausible adverse events may be treated, usually requiring minimal intervention (e.g. anti-histamine).
B. Considering triggers of CSU - (see. Line 215) there are many concerns raised by patients and currently it seems that there is much data regarding infections, NSAID and stress while data on food is very limited and thus may not be regarded as such.
Author Response
Point-by-point response to Editors and Reviewers
Date: 10-07-2023
Manuscript Number: viruses-2482563
Title of Article: How infection and vaccination are linked to acute and chronic urticaria: with a special focus on COVID-19
Name of the Corresponding Author: Marcus Maurer
Email Address of the Corresponding Author: marcus.maurer@charite.de
Dear Louies,
We thank you for your review and the opportunity to revise our manuscript “How infection and vaccination are linked to acute and chronic urticaria: with a special focus on COVID-19” (viruses-2482563). We diligently made all of the revisions requested by the reviewers as described in our detailed point-by-point response below and highlighted changes in the document keeping the track changes on. All authors agree with the revised manuscript.
We hope that our revisions are regarded as sufficient so that a re-evaluation of the manuscript may now support its publication in Viruses Journal. Again, we thank you and all reviewers for the positive feedback and suggestions, which helped us to improve our manuscript.
With our best wishes,
Emek and Marcus
Main changes done:
- Changes according to reviewers’ suggestions are carried out
- Changes in Table 2 are done, the table has more detailed data now
- References are revised and 2 references are removed
- Limitations part revised and extended
Reviewer 2
This is a well written comprehensive review of the current knowledge regarding plausible associations between COVID-19 infection and/or vaccination and acute/chronic urticaria.
Reply: We thank the reviewer for their positive evaluation of our manuscript.
2 minor comments
- It may be of important to emphasize that although anti-COVID vaccine may be linked with urticaria and particularly exacerbation of CSU - It is not recommended to withhold further immunization and this plausible adverse events may be treated, usually requiring minimal intervention (e.g. anti-histamine).
Reply: We thank the reviewer for their important suggestion. We included this sentence in the conclusion. Now the text reads as below:
However, it is also important to stress that, although COVID-19 vaccines may be linked with urticaria and exacerbation of CU - It is not recommended to withhold further immunization and this plausible adverse event may be treated, usually requiring minimal intervention (e.g. anti-histamine).
- Considering triggers of CSU - (see. Line 215) there are many concerns raised by patients and currently it seems that there is much data regarding infections, NSAID and stress while data on food is very limited and thus may not be regarded as such.
Reply: We thank the reviewer for their valuable insight. We agree with them, thus we removed ‘’foods’’
(I) Please revise your manuscript according to the referees’ comments and
upload the revised file within 5 days.
(II) Please use the version of your manuscript found at the above link for
your revisions.
(III) Please check that all references are relevant to the contents of the
manuscript.
(IV) Any revisions to the manuscript should be highlighted, such that any
changes can be easily reviewed by editors and reviewers.
(V) Please provide a short cover letter detailing your changes for the
editors’ and referees’ approval.
